# Thermal control of the topological edge flow in nonlinear photonic lattices

**Pawel S. Jung** [1,2,6], **Georgios G. Pyrialakos** [1,6], **Fan O. Wu** [1], **Midya Parto** [1], **Mercedeh Khajavikhan** [1,3], **Wieslaw Krolikowski**[4,5] & **Demetrios N. Christodoulides** [1] ✉

The chaotic evolution resulting from the interplay between topology and nonlinearity in photonic systems generally forbids the sustainability of optical currents. Here, we systematically explore the nonlinear evolution dynamics in topological photonic lattices within the framework of optical thermo-dynamics. By considering an archetypical two-dimensional Haldane photonic lattice, we discover several prethermal states beyond the topological phase transition point and a stable global equilibrium response, associated with a specific optical temperature and chemical potential. Along these lines, we provide a consistent thermodynamic methodology for both controlling and maximizing the unidirectional power flow in the topological edge states. This can be achieved by either employing cross-phase interactions between two subsystems or by exploiting self-heating effects in disordered or Floquet topological lattices. Our results indicate that photonic topological systems can in fact support robust photon transport processes even under the extreme complexity introduced by nonlinearity, an important feature for contemporary topological applications in photonics.

The recently discovered topological phases of matter have introduced new frontiers in fermionic and bosonic systems by exposing a number of intriguing and unconventional physical phenomena. In photonics, the prevailing signature of non-trivial topological order is associated with the emergence of unidirectional edge transport that in turn can sustain itself against backscattering from defects or disorder[1,2]. This effect has thus far been observed in a variety of arrangements, involving photonic crystals, waveguide arrays and coupled cavities, to name a few[3–9]. These systems can exhibit diverse topological properties with a variety of phases, including the archetypical Haldane model with non-zero Chern invariants[10], or more complex Floquet phases with higher dimensionality and gaps that are classified by non-trivial winding numbers[11,12]. Photonics has also provided an accessible route to advance topological theories into different avenues that may involve the presence of non-Hermiticity or nonlinear interactions[13–17].

It has been recently shown in condensed matter physics that effects arising from quantum correlations and interactions can lead to a novel class of phase transitions that further broaden the classification of topological systems[18]. This, in turn, trailblazed a similar path for photonic topological arrangements where topological effects are manifested through optical nonlinearities. It is currently known that topological edge states can emerge in either fully linear systems or persist as edge soliton solutions in environments that can support such nonlinear formations[19–30]. Yet, as of now, the role of nonlinearity in interacting topological photonic models[13] has not been system-atically investigated in the weakly non-linear regime, in domain, where self-organized structures cannot appear such as breathers, vortices, or solitons[14–16,22,31]. In this case, lattices are heavily multimoded and as such the ensuing non-linear energy exchange among modes can be exceedingly complex[32–37]. In this respect, one may pose a series of

[1]College of Optics & Photonics-CREOL, University of Central Florida, Orlando, FL 32816, USA. [2]Faculty of Physics, Warsaw University of Technology, Koszykowa 75, 00-662 Warsaw, Poland. [3]Ming Hsieh Department of Electrical and Computer Engineering, University of Southern California, Los Angeles, CA, USA. [4]Laser Physics Centre, Research School of Physics and Engineering, Australian National University, Canberra, ACT 0200, Australia. [5]Science Program, Texas A&M University at Qatar, Doha, Qatar. [6]These authors contributed equally: Pawel S. Jung, Georgios G. Pyrialakos. ✉e-mail: demetri@creol.ucf.edu

fundamental questions pertaining to the sustainability of the topological edge currents under the influence of multi-wave mixing processes. In other words, how will these "many-body" topological systems asymptotically respond? More importantly, will they reach an equilibrium state, and can this state be externally controlled? Due to the omnipresence of nonlinearity in a number of contemporary settings, like topological laser systems[17,38,39], an answer to these questions could be vital for the field of topological photonics.

In this Article, we derive a thermodynamic formalism capable of describing the underlying non-linear dynamics of photonic topological insulators in the weakly non-linear regime. Our results indicate that, for a given set of initial conditions, the topological system will always maximize its optical entropy and thus reach an equilibrium state at a specific optical temperature $T$ and chemical potential $\mu$. As such it will attain a state governed by a Rayleigh-Jeans distribution in its modal space that is characterized by a positive or negative optical temperature. Given that this statistical distribution is expressed in the eigenfunction space where all topological properties are defined, it enables one to directly monitor the interplay of topology with the thermalization process. In this respect, we find that such a system preserves its topological structure and unambiguously presents a signature of unidirectional transport at the edges. This property reveals an exclusive path for controlling and maximizing the topological edge flow in a nonlinear topological optical system by means of thermodynamics. We examine two exemplary cases, an optical Haldane lattice and a Floquet topological insulator, and demonstrate how different reversible and irreversible processes can be deployed in a variety of settings.

## Results

### Thermalized and prethermalized states in 2D nonlinear topological insulators

To exemplify our approach, let us begin by considering a honeycomb lattice of M single-mode coupled optical elements, with real nearest and complex next-nearest neighbor hopping amplitudes (Fig. 1a) that can in principle vary in time. The optical non-linear dynamics are governed by the following normalized discrete nonlinear Schrödinger equation:

$$i\frac{da_n}{dt} + \Delta_n a_n + \sum_{NN} \kappa_{1k}(t)a_k + \kappa_2 \sum_{NNN} a_q e^{-i\phi} + |a_n|^2 a_n = 0 \qquad (1)$$

where, $a_n$ represent the optical field amplitude on site $n$ and t is the time variable. Each element is directly coupled to its three nearest neighbors with hoping amplitudes $\kappa_{1k}$ (for $k = 1, 2, 3$) as well as to its next-nearest neighbors (NNN) (over sum $q$) with hoping amplitudes $\kappa_2$ (Fig. 1a). The phase factor $\phi$ is associated with the NNN exchange and can be positive or negative for counterclockwise or clockwise hoping, respectively. An energy offset $\Delta = (\Delta_A - \Delta_B)/2$ between the two sublattices A and B can also be used to break the spatial inversion symmetry. Here, a Kerr nonlinearity is introduced in the arrangement through the last term in Eq. (1). In what follows, we focus on a Haldane lattice with $\kappa_1 \neq 0, \kappa_2 \neq 0$ and no time dependence. A Floquet topological case with $\kappa_2 = 0$ and $\kappa_1$ time dependent may also be considered. In either case the nonlinear dynamics are expected to thermalize in a similar way if we ignore self-heating effects (for fast enough periods of the drive) which will be examined later.

In general, the nonlinear dynamics associated with Eq. (1) represent an utterly complex problem. For example, under non-linear conditions, a system initially excited at a state that maximally couples to the topological edge modes, might inevitably allow energy to leak into the bulk, and as a result, will not conserve its unidirectional light flow. Here, we focus on the weakly non-linear regime where high-power non-ergodic structures, like edge soliton formations, cannot manifest. We tackle this problem through statistical mechanics[40], in a phase space defined by the linear modes of the system when two conserved quantities or invariants are at play[41]. In a conservative arrangement, the first constant of the motion is given by the norm corresponding to the optical power $P = \sum_j |c_j|^2$, where $c_j$ denotes the occupancy strength of the mode with eigenvalue $\varepsilon_j$. The second invariant corresponds to the effective internal energy, given by the linear part $U = -\sum_j \varepsilon_j |c_j|^2$ of the total Hamiltonian $H = U + H_{NL}$, where $H_{NL} = 1/2\sum_j |a_n|^4$ for the Kerr-type nonlinearity. To guarantee that the optical energy remains invariant

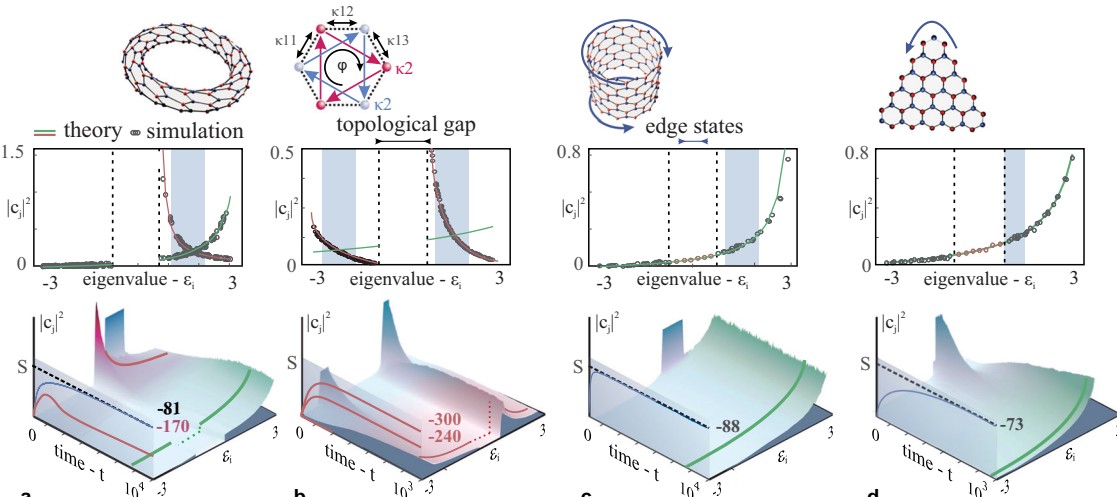

**Fig. 1 | Thermalization of the nonlinear Haldane lattice. a** A nonlinear Haldane lattice ($\kappa_2 = 0.2, \varphi = \frac{\pi}{2}, \Delta = 0$) is excited with a uniform population amongst all supermodes within the gray shaded region. During evolution it develops a prethermal state, indicated by the red region (bottom panel), with a RJ distribution predicted by accounting only the upper modal group. The entropy of this subsystem (red curve) reaches a local maximum and then decreases when coupled to the lower band. Eventually the system reaches global equilibrium (green region) by maximizing its total entropy (blue curve). **b** When excited at both modal groups, the same lattice prethermalizes into two separate local quasi-equilibrium states with the same temperature but different chemical potentials. In this case, the internal energy is rapidly exchanged while the total power is conserved within each modal group. A truncated ribbon (**c**) and triangular (**d**) configuration will thermalize without the manifestation of prethermalization due to the presence of the topological edge states that bridge the band gap.

during propagation, we investigate this system exclusively in the weakly nonlinear regime where the linear part of the energy ($U$) dominates the nonlinear component ($H_{NL}$). To do so, we appropriately control the optical power level so as the normalized Kerr nonlinearity in Eq. (1) is weak. Under the aforementioned conditions, one can show that the "thermalized" modal occupancies, (for all modes, including the topological edge states), obey a Rayleigh-Jeans law, i.e. $|c_j|^2 = -T/(\varepsilon_j + \mu)$[42–45], a distribution that maximizes the underlined optical entropy $S = \sum_j \ln(|c_j|^2)$ (see Supplementary Note II). In addition, the thermodynamic extensive variables ($U, M, P$) are related to the intensive quantities $T$ and $\mu$, through a global equation of state $U - \mu P = MT$, where $M$ is the total number of modes[42]. It is important to note that the effective temperature $T$ and chemical potential $\mu$ are always uniquely determined from the initial excitation conditions of the lattice, i.e. from $U$ and $P$[46] (see Supplementary Note III).

We begin our analysis by investigating the thermalization process in the nonlinear Haldane lattice of Eq. (1) comprising $M = 200$ sites (with an equal number of supermodes) in the topologically non-trivial ($\kappa_2 = 0.5$) phase, with zero detuning ($\Delta = 0$). We employ a torus geometry which can host a collection of modes from the bulk spectrum (for all discrete momenta that satisfy the periodic boundary conditions) and can therefore serve as a proper representation of bulk dynamics. Subsequently, this can provide a juxtaposition to the more realistic case of a truncated system which involves a complete set of topological edge states. We examine two examples by exciting in the first case only the upper band group with a uniform occupation strength amongst modes $j \in \langle 120 - 150 \rangle$ (Fig. 1a) and in the second case both band groups at modes $j \in \langle 60 - 90 \rangle$ and $j \in \langle 110 - 140 \rangle$ (Fig. 1b). To identify the complete asymptotic behavior of these two systems we observe their dynamics over a long period of time, long enough for an equilibrium state to be established. In Fig. 1a, b, the green curves correspond to the final RJ distribution as predicted by the internal energy ($U$) associated with each excitation (in both cases power $P = 6$). However, the presence of the topological gap does not only slow down the thermalization process but also promotes the development of a temporary prethermal state, outlined with a red curve. This state is distinguished by a local RJ distribution and a new temperature defined by accounting exclusively the respective modal group, associated with either Chern number $\mathscr{C} = 1$ or $\mathscr{C} = -1$, in the calculation of the internal energy $U$. Seemingly, the life-time of this state is directly correlated to the size of the topological gap, which can be regulated by the magnitude of the coupling $\kappa_2$. In the particular case of Fig. 1b the two modal groups interact by exchanging power at a much smaller rate that energy, ultimately reaching the same optical temperature but different chemical potential. Eventually the system will settle into the global RJ curve (in green) in a similar fashion to Fig. 1a.

A similar prethermalization process can also be observed in a trivial system where the sublattice symmetry is broken via the parameter $\Delta$. In this case, however, the topologically trivial and non-trivial phases are very different. In general, the gap size of a trivial system does not depend on its configuration (torus or truncated) and therefore the system is expected to thermalize in a similar manner, if all other conditions are met. Conversely, in a non-trivial topological lattice, the thermalization process is profoundly different depending on whether it unfolds in a torus or a truncated system. In Fig. 1c, d, we replicate the non-trivial case of Fig. 1a, with the same lattice parameters, but instead in a ribbon as well as a triangular topological configuration. The helical edge states that emerge in these arrangements bridge the band gap and lead to a rapid thermalization as predicted by a global RJ distribution, without being preceded by a prethermal state. This is in stark contrast to the bulk dynamics observed in a torus lattice (Fig. 1a), an effect that is directly attributed to a thermal version of the bulk-edge correspondence principle.

## Topological currents at thermal equilibrium

Having studied the prethermalization and thermalization dynamics in these nonlinear topological systems, we would like to next identify their impact on topological currents. As we will see, under thermal equilibrium conditions, the flow of an optical unidirectional edge current still persists, in spite of the extreme complexity resulting from the presence of nonlinear interactions. Given the fact that the relative phases between modes vary in a stochastic fashion, the average discrete current density $J_n$ on a lattice site $n$ at equilibrium, can be expressed as an incoherent sum of the partial currents $\Omega_{n,j}$ resulting from each mode:

$$J_n = \sum_j^M |c_j|^2 \Omega_{n,j} = \sum_j^M \frac{-T}{\varepsilon_j + \mu} \left\langle \Psi_j | \hat{J}_n | \Psi_j \right\rangle \qquad (2)$$

where $|\psi_j\rangle$ are the lattice eigenvectors and $\hat{J}_n$ denotes the local current density operator defined by $\hat{J}_n = \sum_m \vec{\delta}_{n,m}(t_{n,m} - H.c.)$[47]. Here, $t_{n,m}$ and $\vec{\delta}_{n,m}$ are the coupling coefficients and the vector displacement between any site $n_i$ and $n_j$, respectively.

We begin by examining a triangular Haldane configuration with $N = 61$ sites and a zig-zag terminated edge when $\kappa_2 = 0.2$ and $\phi = \frac{\pi}{2}$. This system will always settle into the predicted RJ distribution which allows one to calculate from Eq. (2) the corresponding average contribution $\Omega_{n,j}$ to the total current $J_T = \sum_{n=1}^N J_n$ circulating in the lattice—a discrete version of the line integral $\oint_C \mathbf{J} \cdot \mathbf{dl}$, where $C$ represents the outer contour of the topological lattice. In Fig. 2a we observe the evolution of the total current for $T = 0.19$ and $\mu = -4.85$ as well as the development of the total current contributions from the edge states $J_E$ (which establish a unidirectional propagating channel at the boundary) and the bulk modes $J_B$ (which contribute only through local variations). A closer look into the non-equilibrium dynamics reveals that at the beginning ($t \approx 0$) the total current is largely dominated by the edge state contribution. During evolution, the optical power is "thermally" redistributed between edge and bulk states, thus establishing a competition between $J_E$ and $J_B$. Surprisingly and counterintuitively, as the time progresses, while $J_E$ tends to initially decrease, it eventually stabilizes to a new average value thus indicating that the edge current is robust despite thermalization as induced by nonlinear multi-wave mixing effects. Once the optical entropy reaches its maximum value, the average light transport changes direction (thus flipping the sign of the current) and eventually attains its analytically predicted negative value. Next, we calculate the average unidirectional current density at each site $n$ of the lattice by examining the same system at various temperatures (Fig. 2b). From here, it becomes apparent that the two contributions to $J_T$ are always associated with opposite helicity. At $T = \infty$ the two parts completely negate each other resulting in a net zero average current while as temperature decreases, $J_B$ becomes the dominant current contribution and $J_T$ attains a finite value.

Having investigated various cases with specific optical temperatures we must next explore the complete parameter space of the system in order identify regimes where the total current is enhanced. To address this issue, we explore the parameter space of Eq. (1) by varying the internal energy $U$ associated with the initial excitation while keeping the total power $P$ constant. Figure 2c illustrates the total average current amplitude summed over all sites in the topological Haldane lattice when the phase $\varphi$ varies in the range $-\pi \leq \varphi \leq \pi$ and $\Delta = 0$, $\kappa_1 = 1$, $\kappa_2 = 0.2$. The absolute maximum value is reached at a relatively low temperatures where most of the power will occupy the lower or upper energy states. Interestingly, at a positive temperature ($T \approx 0.1$), the total current displays two extrema at $\varphi \approx \pm \frac{3\pi}{4}$, while in the negative temperature regime ($T \approx -0.1$) these extrema shift at $\varphi \approx \pm \frac{\pi}{4}$. By keeping, instead, the parameter value of $\varphi$ constant, we can also identify a particular point, at $\varphi = \frac{\pi}{2}$, where the temperature dependence of the current displays a strictly symmetric behavior around

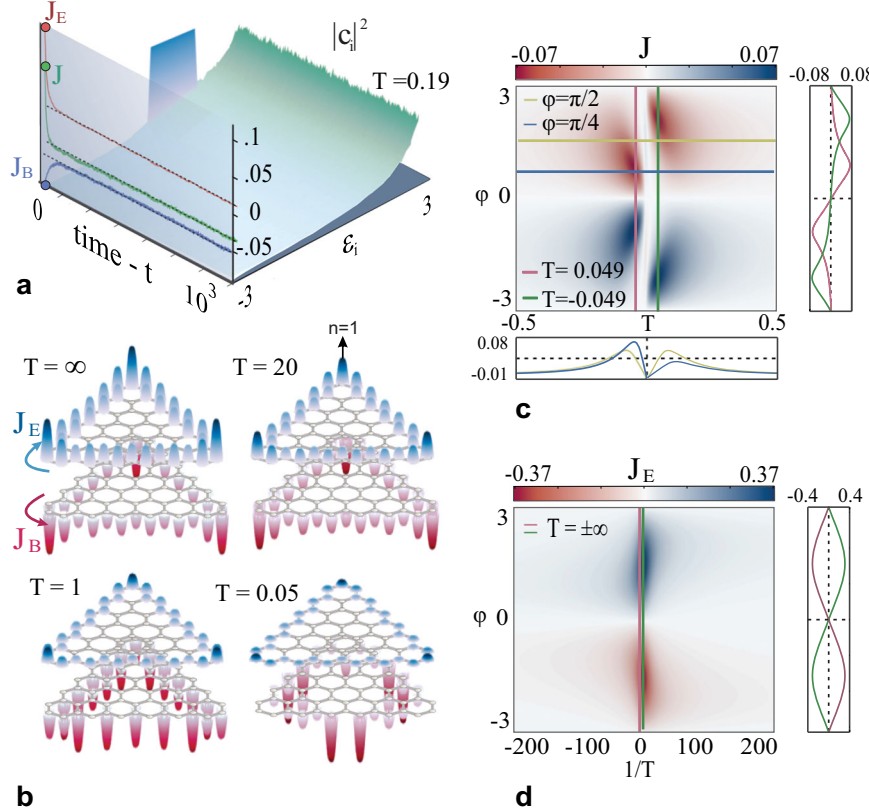

**Fig. 2 | Topological currents at equilibrium.** When a topological system reaches an equilibrium state, the magnitude of the total current will be equal to the sum of the contributions from all modes with occupancies $|c_i|^2$ as dictated by a RJ distribution. **a** Time evolution of the modal occupancies and topological currents for a Haldane lattice with $T = 0.19$ and $\mu = -4.85$ ($J_E$ red−contribution from edge modes, $J_B$ blue−contribution from bulk modes, $J$ green−total current). **b** At lower temperatures the contribution to the topological current from the edge states (upper part−blue) decreases while the contribution from the bulk states (lower part −red) rises. Average (**c**) total and (**d**) edge state current summed over all sites in a Haldane lattice for variable temperature $T$ and topological flux $\varphi$. The outer panels corresponds to the colored lines of the respective maps.

$T = 0$. Conversely, Fig. 2d illustrates only edge states contribution which, evidently, assumes its maximum value at infinite temperatures, when power equipartition among modes takes place. At $T \to \pm\infty$ the amplitude varies periodically over $\varphi$, reaching a global maximum at $\varphi = \frac{\pi}{2}$, as shown in the same figure.

**Maximizing the topological edge flow via thermal processes**

The topological current maps of Fig. 2 reveal a clean path for maximizing the topological currents through thermodynamic processes that can induce a shift in the optical temperature. A conventional way to perform this is via external means, i.e. using an optically hotter secondary lattice (which can be non-topological) to cool the primary system under investigation. Alternatively, here, we explore the prospect for nonlinear cross-interactions between two distinct species in a topological Haldane system that can behave as two different optical subsystems (with different temperatures). In this respect, the two species could be for example the left ($a_n^L$) or right-handed ($a_n^L$) circular polarizations. Each polarization obeys separately Eq.(1) while interacting with each other via cross-phase modulation terms ($\gamma |a_n^L|^2 |a_n^R|^2$). In this case, we are interested in maximizing the total current using the temperature curve of Fig. 2c. Figure 3a illustrates these results for a triangular configuration both in the presence or absence of the cross-phase modulation terms. In both cases, the modes with eigenvalues in the range $1.3 \le \varepsilon \le 2.97$ and $-1.07 \le \varepsilon \le 1.49$ are uniformly excited with the same total power in the left and right-handed polarization. As one would expect, in the absence of cross-phase modulation ($\gamma = 0$), the two polarizations preserve their internal energy and optical power, ultimately reaching different equilibria, $T = 0.043$ and $\mu = -2.87$, for the left-

handed, and $T = 0.19$ and $\mu = -4.85$ for the right-handed. Eventually, the currents for both subsystems, will reach their theoretically predicted points on the J–T curve (Fig. 3a). When the cross-phase modulation is engaged ($\gamma = 2$), the two polarizations exchange only their energies and as a consequence they reach a common temperature ($T = 0.083$). As a result, the magnitude of the equilibrium current shifts at the top of the J–T curve, located between the two initial points, assuming its maximum value in agreement with theoretical predictions.

Next, we consider methodologies to optimize the current flow $J_E$ (contributed by the topological edge states) which as we saw before (Fig. 2d.) it reaches a maximum at (positive or negative) infinite temperatures. As we show below, this can be achieved by means of an irreversible heating process, i.e., through a steady absorption or removal of optical energy at negative and positive temperatures respectively (which in our lattice system corresponds to $U = 0$). In such a case, the entropy will increase in time, eventually reaching its maximum in the S(U) curve (Fig. 3b). Such an irreversible mechanism can be accomplished via either introducing a time-varying disorder in a Chern topological system (such as a Haldane lattice) or in periodically driven Floquet topological systems. For the latter case, we consider a honeycomb Floquet lattice with time-periodic coupling variations (with a clockwise rotating phase difference). It is generally known in classical thermodynamics that time-periodic systems at positive temperatures tend to continuously absorb heat until they reach an infinite temperature state[48,49], a feature that we can successfully replicate within the framework of optical thermodynamics. In the negative temperature regime, the converse is true. Here, we simulate both these

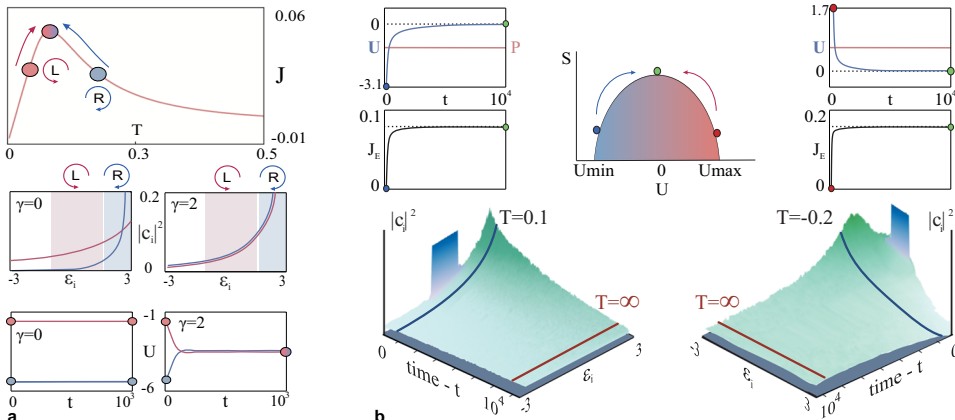

**Fig. 3 | Thermal control of the topological edge flow. a** Maximizing the topological current $J_T$ via nonlinear cross-phase interactions between left (L) and right-handed (R) circularly polarized light. The results were obtained for a triangle configuration, with $M = 61$ sites, in the nonlinear Haldane lattice with $\kappa_1 = 1, \kappa_2 = 0.2, \varphi = \frac{\pi}{2}$. At $t = 0$, both polarizations have the same optical power $P_R = P_L = 2.8$ but different internal energies $U_L = -1.77$ and $U_R = -5.4$. The upper panel depicts the evolution of the two subsystems for $\gamma = 2$ which trace the red line corresponding to the theoretical $J_T$-Temperature curve, towards a common T at equilibrium that maximizes the two currents. The middle panels depict the initial (shaded regions) and equilibrium (line plots) modal occupancies for the left (red) and right-handed (blue) polarizations for $\gamma = 0$ (left panel) and $\gamma = 2$ (right panel).

The bottom panels show the conservation/exchange of the internal energy ($U$) between polarizations for the case with/without cross-phase modulation. **b** Self-heating of a disordered (left) and a Floquet (right) topological lattice. The middle panel depict schematically the entropy at $t = 0$ (red and blue dots) as well as at equilibrium (green dot) after self-heating has occurred. The top-edge panels show the gain and lose of the internal energy during self-heating until a zero value is reached, where the edge current maximizes its value. During evolution, both lattices first prethermalize at $T = 0.1$ and $T = -0.2$ (for a Haldane and Floquet lattice, respectively) and then adiabatically shift their temperatures until they reach a $T = \infty$ state.

cases by exciting a uniform set of eigenvalues with a total power of $P = 2.8$ and internal energy $U = -3.1$ ($T = 0.1$) and $U = 1.7$ ($T = -0.2$) for the Haldane and the topological Floquet lattice, respectively. Figure 3 illustrates the adiabatic process of energy gain in the disordered Haldane lattice which allows the self-optimization of the topological current through reshuffling of the optical power in the linear modes due to random multi-wave mixing phenomena. In a similar vein, the topological Floquet lattice optimizes its current flow by driving the system from a prethermalized state (blue line) into a state where equipartition occurs (red line), by continuously shedding excess energy. In the Floquet case, the speed of time modulation generally dictates the overall lifetime of the pre-thermalized state (Fig. 3).

## Discussion

In this work we have systematically investigated the thermodynamic evolution of nonlinear photonic topological systems, where several intriguing phenomena before and during thermal equilibrium have been identified. In this regard we found that the presence of prethermal states beyond the topological phase transition point. Our analysis reveals for the first time that, under thermal equilibrium conditions, optical unidirectional edge currents still persist in such topological systems, in spite of the extreme complexity introduced by thermalization through multi-wave mixing nonlinear interactions. Most importantly, we developed a self-consistent theoretical framework within which one can precisely predict the magnitude of the thermalized currents in nonlinear photonic topological insulators from arbitrary initial conditions. In this respect this methodology can be effectively deployed to control and maximize the unidirectional power flow in the topological edge states.

The results presented herein can be relevant to a broader class of bosonic and fermionic topological configurations that can be appropriately described within the semiclassical domain. These may include for example mean-field models of interacting many-body bosonic systems such as Bose-Einstein condensates in optical (trapping) lattices governed by the Gross-Pitaevski equation, or paired fermionic

arrangements that also exhibit condensation. In this respect, our findings are general and can be readily adapted to different topological platforms that allow one to dynamically control the on-site potentials in the weakly interacting regime.

Possible future direction of interest will be to expand the optical thermodynamic theory into different regimes in order to more comprehensively describe topological settings in the presence of high nonlinearities. In the weakly nonlinear regime, a thermodynamic treatment is possible due to the emergence of ergodicity. Beyond this domain, self-organized structures can appear such as breathers or vortices, or in the case of topological systems, edge solitons. For example, in topological systems, edge solitons can form at the boundaries of the arrangement that retain the topological robustness of regular edge states. Deciphering the statistics of fluctuating edge solitons will unlock new possibilities in controlling the formation of such topologically robust structures via the manipulation of their thermodynamic phase transitions. In this respect, the thermal collapse of edge solitons into a gas of photons (via a transition to the weakly nonlinear regime) may provide new pathways in controlling the thermalization of photonic topological lattices. Finally, our findings regarding prethermalization, could provide a new pathway for controlling the thermal evolution of systems with multiple topological gaps by employing different configurations with periodic or truncated boundaries.

Quite recently, optical thermalization effects have been observed for the first time in optical multimode fibers[50,51] and time-synthetic lattices[52]. In principle, photonic topological lattices can be inscribed in silica glass[4,9,53] or can be implemented in time-synthetic dimensions. Given the long evolution times required to observe thermalization effects, a possible platform to observe such thermalization phenomena in 2D nonlinear topological lattices could be time-synthetic arrangements along the lines discussed in Refs. 54,55. These 2D systems are typically implemented involving four fiber loops and allow hundreds of exchange times—that is more than enough to observe effects along the lines discussed in our paper. In general, ribbon like geometries with periodic boundary conditions can also be implemented in 2D time-synthetic lattices.

## Reporting summary

Further information on research design is available in the Nature Research Reporting Summary linked to this article.

## Data availability

The data that support the findings of this study are available from D.N.C. upon reasonable request.

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

## Acknowledgements

This work was partially supported by ONR MURI (N00014-20-1-2789), AFOSR MURI (FA9550-20-1-0322, FA9550-21-1-0202), National Science Foundation (NSF) (DMR-1420620, EECS-1711230,ECCS CBET 1805200, ECCS 2000538, ECCS 2011171), MPS Simons collaboration (Simons grant 733682), W. M. Keck Foundation, US–Israel Binational Science Foundation (BSF: 2016381), US Air Force Research Laboratory (FA86511820019), DARPA (D18AP00058), Office of Naval Research (N00014-19-1-2052, N00014-20-1-2522), Army Research Office (W911NF-17-1-0481), the Polish Ministry of Science and Higher Education (1654/MOB/V/2017/0) and the Qatar National Research Fund (grant NPRP13S-0121-200126). G.G.P. would like to acknowledge the Bodossaki foundation.

## Author contributions

G.G.P. and P.S.J. contributed equally to this work, initiated the idea, and performed the theoretical calculations and simulations. M.P. assisted on the theoretical analysis. Finally, F.O.W, M.K., W.K., and D.N.C. discussed the results and contributed in preparing the manuscript.

## Competing interests

The authors declare no competing interests.

## Additional information

**Peer review information** *Nature Communications* thanks Other anonymous Reviewer(s) to the peer review of this work. Peer review Reports are available.

