## [Peer Review File · Nature Communications]

REVIEWER COMMENTS

Reviewer #1 (Remarks to the Author):

The manuscript “Thermal control of the topological edge flow in nonlinear photonic lattices” by P.S. Jung et al is devoted to the study of the average current in a topological photonic system in a weakly-interacting regime, where the populations of linear eigenstates can be described with an equilibrium distribution function. The authors claim that in this case it is possible to observe a non-zero current associated with the topological edge states.

This is an interesting work, making a kind of a bridge between topological photonics and “topological electronics” (electronic topological insulators) by applying the methods of statistical physics to non-linear photonic systems. I think that this work has a potential to be published in Nature Communications. However, it needs to be made clearer.

Overall, a general qualitative conclusion needs to be provided along the following lines. The thermalization of the photonic system makes it similar to the electronic topological insulators, except that the statistics is different. Is it true, that the main qualitative conclusions remain more or less the same? Or there are some qualitative differences, for example, linked with the presence of a Fermi level? In Fermionic systems, usually only the levels close to the Fermi energy (partially empty or partially filled) can contribute to the conductivity and the currents. Maybe here it is different? What are the consequences?

The thermalization of photons (or waves in general) in a non-linear system occurs only under certain well-defined conditions. The authors write “under aforementioned conditions”, but unfortunately they are not specified explicitly. It would be really useful for the work to indicate them clearer, for example, using the conditions for weak turbulence from Ref. 38. Unfortunately, I do not have the access to this work myself, but the authors probably do, since they cite it. The abstract of Ref. 38 states “the reader can gain a perspective on the ingredients important for the realization of the various equilibrium spectra, thermodynamic, pure Kolmogorov and combinations thereof”. What are the ingredients important for the realization of equilibrium spectra? And what can these spectra be, potentially?

To put the work in the context of interacting topological photonics, it is probably important to cite another, more recent, review on non-linear topological photonics, Appl Phys Reviews 7, 021306 (2020). It is interesting to note that the quantum fluid of lights (such as exciton-polaritons) provide a broader set of classes of solutions possible in non-linear systems. The authors restrict their discussion to two possibilities: solitons or weakly-interacting thermalized system. Other situations are possible and studied. In particular, it is possible to study a Bose-Einstein condensate of polaritons (thermalized in the ground state) or an out-of-equilibrium quantum fluid. The weak excitations of the condensate (bogolons) can exhibit topologically non-trivial bands with edge states. The topology can be induced by spin-anisotropic interactions of the quantum fluid itself (no gap in the linear case). Yet another possibility is to study the behavior of vortices (and not solitons) in a quantum fluid filling the system.

The authors point out the role of the edge state in the suppression of the prethermalization. This role can be expected to be vanishing for an infinite system size. Is this true? The authors should probably comment on this.

The authors need to explain qualitatively why their model predicts a non-zero bulk current. All bulk modes are symmetric with respect to positive and negative current directions, and therefore for any temperature the average bulk current can be expected to be zero. Probably this is some not very obvious feature of the system or of the model that I do not understand. In the electronic systems, the current in the bulk Landau levels is zero. Why is it different here? Why there is some negative contribution into the current from the bulk modes along the edges in Fig. 2?

In the supplementary, the authors state that “during prethermalization, the two bands exchange optical energy much faster than power”. Are there any explanations or arguments in favor of this statement?

Some small typos should be corrected, including the word « eigenvalue » in Fig 1.

Reviewer #2 (Remarks to the Author):

This is a report on the manuscript entitled "Thermal control of the topological edge flow in nonlinear photonic lattices" that is being considered for publication in Nature Communications. The manuscript presents numerical results elaborating the complex interplay of optical nonlinearity in topologically nontrivial materials. The system consists of tight-binding honeycomb lattices, e.g., a static case of Haldane model with complex-valued couplings and a time-periodic Floquet topological insulator. The central aspect is the "long-time" evolution of optical states in the topological lattices within the framework of optical thermodynamics. Thermalisation in the "optical sense" and the development of prethermal states have been studied. The authors show that the topological systems can support robust transport in the presence of complicated nonlinear processes.

Topologically robust transport in the linear domain has been well studied over the past decade. On the other hand, nonlinear topological photonics is a new research direction. Very recently, the formation of Floquet solitons and quantised nonlinear transport has been experimentally observed in topological lattices, see Science 368, 856 (2020) and Nature 596, 63 (2021) [The authors may cite these papers].

The manuscript is well written in general, and the results are novel.

I believe that this work deserves publication in a decent journal. However, I think a general reader would like to know about the challenges in experimentally probing such effects. Apparently, probing optical

states for an extremely long evolution time (or propagation distance) is difficult using photonic topological lattices. Additionally, both periodic and open boundary conditions are used for analysing the thermalisation process -- it would be nice to highlight how the periodic boundary condition can be realised in experiments.

Given the high impact of the journal, it would be nice to elaborate further on the most significant aspect of their results and the perspectives for new research that will be enabled by it.

Reviewer #3 (Remarks to the Author):

Jung et al. study theoretically the equilibration in nonlinear, topological model systems. They have in mind optical implementations using laser-written waveguides, for instance, where nonlinearity comes about because of the Kerr effect. Next-nearest neighbour hopping of the Haldane kind or periodic driving make the system topologically nontrivial.

The authors employ the analogy with thermodynamics where the eigenmodes assume the role of energy eigenstates, their occupation determines the internal energy, and the optical power corresponds to the particle number. The entropy (to be maximised) is then defined by the usual $\ln W$, where W is the number of ways the energy may be distributed among the possible modes. All this allows to define an optical temperature that characterises the equilibrium state the system reaches after sufficiently long time propagation.

The authors find that a band gap hinders the overall equilibration, which is expected because the equilibration among the modes can only work within a band but not across band gaps. In finite topological systems, however, there are edge states present that connect bands in a continuous fashion so that equilibration takes place for the entire system. The authors find very good agreement with the theoretically predicted Raleigh-Jeans distribution.

Next, the authors study whether the edge currents (i.e., the main and exciting feature of topologically non-trivial systems) survive in such nonlinear systems in "thermal" equilibrium. As expected, the edge current decreases over time but the fact that it does not approach zero asymptotically but a finite value is very interesting. Finally, the authors investigate certain strategies to optimise the edge flow despite the nonlinearity.

I find the work very interesting, the methods applied are standard but appropriate to study the complex equilibration dynamics in a systematic fashion. The findings are sound and relevant to the community working in the field of topological photonics, but also beyond because the kind of nonlinearity considered also appears in mean-field descriptions of interacting many-body systems with contact-type interactions (e.g., the Gross-Pitaevskii equation). I recommend publication in Nature Communications after the authors have addressed the following issues.

1. In the text, on page 6, the authors state that at $T=\infty$ the currents J_E and J_B “negate each other resulting in a net zero average current”. However, I do not see this in Fig. 2a where J_T is not approaching zero for large t . Moreover, J_E (red) + J_B (blue) is not equal to J_T (green) in Fig. 2a.

2. How did the authors separate edge modes from bulk modes in the first place? Figures 1c and 1d suggest that they consider edge modes as the modes for energies where - without edges - there would be the band gap. On the other hand, one should look at the localization of the modes close to the edges, and then, by definition, there can be no contribution from bulk modes to the edge currents. The authors should clarify their choice.

3. I had a hard time to go through (and understand) all panels in Fig. 3. First of all: Is the time axis in 3b, lower right plot, in the right direction? $T=\infty$ is at $t=0$ there. Further, in the caption, the authors write for the upper plot in 3a about “ $\gamma=0$ and $\gamma=2$, respectively”. Hence, I first was looking for two curves until I understood that one moves along the theoretical, red line for nonvanishing γ and does not for $\gamma=0$. Further, the authors write that “the initial and equilibrium modal occupancies” are depicted in the middle panel. It is clear that the lines are the equilibrium distributions. The fact that the shaded areas indicate the initial occupancies follows from what is written in the main text but not from the caption. The authors should improve on the caption and/or the figure itself.

4. There is a small ϕ in Fig. 2b at the axes but a capital ϕ in the text.

5. It should read “loss of the internal energy” in the figure caption of Fig. 3, not “lose of the internal energy”.

Response to Reviewers

We would like to thank all our reviewers for carefully reading our work and for their constructive comments. In this letter, we provide our response to our referees' remarks in the order they appeared in their reviews. Following our reviewers' suggestions, we have appropriately revised our manuscript and the supplementary section.

We believe that these revisions have substantially improved the quality of our work and we hope that our paper is now suitable for publication in Nature Communications.

Reviewer 1

We would like to thank our reviewer for finding our results interesting for in Nature Communication. We would also like to thank him/her for the constructive comments and suggestions. In response to our reviewer's comments, we have added a discussion section (page 7) in our manuscript and we have expanded the Supplementary materials (Section 4).

Reviewer's comments: Overall, a general qualitative conclusion needs to be provided along the following lines. The thermalization of the photonic system makes it similar to the electronic topological insulators, except that the statistics is different. Is it true, that the main qualitative conclusions remain more or less the same? Or there are some qualitative differences, for example, linked with the presence of a Fermi level? In Fermionic systems, usually only the levels close to the Fermi energy (partially empty or partially filled) can contribute to the conductivity and the currents. Maybe here it is different? What are the consequences?

Response: Indeed, as our reviewer indicated the thermalization dynamics in the photonic system considered in our work has similarities with electronic topological insulators, in spite of the fact that the "carrier" statistics are fundamentally different. This should have been anticipated given that in our case we deal with a classical bosonic system while in solid state these statistics are governed by a Fermi-Dirac distribution. For instance, in our case, at zero temperature the system can condense in the ground state while in fermionic arrangements all occupied states are determined by the Fermi level. As indicated above, there are also similarities when it comes to the antagonistic interplay between bulk and topological edge states under the influence of thermalization effects, but unlike solid state, the chemical potential (corresponding to the Fermi level) does not seem to ultimately dictate the edge current response of the thermalized non-linear insulator. In this regard, we show that the currents in photonic topological systems emerge from the complete Rayleigh-Jeans distribution rather than from specific energy levels. Of interest would be to investigate if there is a relationship between the "conductivity" of the optical edge currents with the optical thermal conductivity (Commun Phys 3, 216 (2020)) in nonlinear photonic topological insulators that is analogous to that expressed by the Wiedemann–Franz law.

In order to put our work in a broader perspective, we expanded the Discussion section in the revised manuscript.

Reviewer's comments: The thermalization of photons (or waves in general) in a non-linear system occurs only under certain well-defined conditions. The authors write "under aforementioned conditions", but unfortunately they are not specified explicitly. It would be really useful for the work to indicate them clearer, for example, using the conditions for weak turbulence from Ref. 38. Unfortunately, I do not have the access to this work myself, but the authors probably do, since they cite

it. The abstract of Ref. 38 states “the reader can gain a perspective on the ingredients important for the realization of the various equilibrium spectra, thermodynamic, pure Kolmogorov and combinations thereof”. What are the ingredients important for the realization of equilibrium spectra? And what can these spectra be, potentially?

Response: As indicated in references xyz, the thermalization of optical waves in a nonlinear multimode (many-state) photonic system is a universal phenomenon that arises under very general conditions. All is needed to induce thermalization is a weak nonlinearity that can chaotically reshuffle the optical power among modes thus allowing a system to explore its phase space in a fair manner so as to be ergodic. In fact, the onset of the Rayleigh-Jeans (RJ) distribution (in the presence of two invariants) reflects on these very principles. In our case, the two invariants are: the total optical power P (total number of particles) and the internal energy of the system which involves a linear part $U = -\sum_j \epsilon_j |c_j|^2$ in the total Hamiltonian $H = U + H_{NL}$ and a nonlinear component $H_{NL} = 1/2 \sum_j |a_n|^4$ accounting for the Kerr-type nonlinearity. In all cases to ensure the invariance of the optical energy U we operate the system in the weakly-nonlinear regime where the quadratic term (U) can efficiently dominate over the quartic (H_{NL}). To operate in this regime we appropriately control the magnitude of the total optical power by involving a normalized Kerr nonlinearity in Eq. (1). As recent studies have shown Ref.42,45 (in revised manuscript), in general, the RJ thermalization always ensues in order to maximize the entropy of the system in accord with the second law of thermodynamics. Alternatively, one can interpret the RJ law from the perspective of a turbulence theory, ref 43 (former Ref. 38) in our paper, which we also attach in our reply for the convenience of our reviewer. We would like to emphasize that the turbulence perspective is applicable when the number of modes tends to infinity while the entropic [42,45] can be deployed when the number of states is large but finite as in the case discussed in our paper. For this reason, the equilibrium spectra (Kolmogorov) play no role in our case.

To elucidate these aspects, we have modified our manuscript, please see page 3, and line 90-92.

Reviewer’s comments: To put the work in the context of interacting topological photonics, it is probably important to cite another, more recent, review on non-linear topological photonics, Appl Phys Reviews 7, 021306 (2020).

Response: We would like to thank the reviewer for bringing this work to our attention. We now cite this article in the revised manuscript as ref. 13.

Reviewer’s comments: It is interesting to note that the quantum fluid of lights (such as exciton-polaritons) provide a broader set of classes of solutions possible in non-linear systems. The authors restrict their discussion to two possibilities: solitons or weakly-interacting thermalized system. Other situations are possible and studied. In particular, it is possible to study a Bose-Einstein condensate of polaritons (thermalized in the ground state) or an out-of-equilibrium quantum fluid. The weak excitations of the condensate (bogolons) can exhibit topologically non-trivial bands with edge states. The topology can be induced by spin-anisotropic interactions of the quantum fluid itself (no gap in the linear case). Yet another possibility is to study the behavior of vortices (and not solitons) in a quantum fluid filling the system.

Response: As our reviewer indicated, there are indeed many excitations that can emerge in interacting bosonic systems, including systems with non-trivial topologies. This is the case, for instance, with exciton-polaritons, Bose-Einstein condensates and cold atoms that may also support excitations like breathers,

vortices, or soliton states. Unlike bogolons that are typically governed by a Ginsburg-Landau equation (a dissipative equation), our thermodynamic framework can be broadly utilized to describe all weakly-nonlinear multimode systems that are more peripheral to the Gross-Pitaevski equation. In this case, a thermodynamic treatment is possible in the weakly-nonlinear regime, as pointed out in our previous response, due to the emergence of ergodicity. Beyond this regime, self-organized structures can appear such as vortices or in the case of topological systems, edge solitons. The development of a thermodynamic theory for these regimes is currently an open topic in optical thermodynamics. Our group has very recently made progress towards this direction and we hope to present these results in the near future. For now, we included a brief discussion on these aspects in the revised manuscript.

To elucidate these aspects, we have modified our manuscript, please see page 2, and line 42-43.

Reviewer’s comments: The authors point out the role of the edge state in the suppression of the prethermalization. This role can be expected to be vanishing for an infinite system size. Is this true? The authors should probably comment on this.

Response: A truly infinite system has a continuous Bloch-Floquet spectrum in the k_x - k_y space. In the topologically non-trivial phase, there will be two bands separated by a gap that is almost equal to the size of the gap of the torus configuration of Fig.1a. In this case, the periodic torus configuration is a good representation of the bulk dynamics, as discussed in the main manuscript.

On the other hand, there is a different aspect when considering this question. This is in regard to the role of the edge states as the system size increases towards to infinity, but still remaining finite. In this case, the edge states will always be present but their role in suppressing prethermalization will be diminished as the system size increases. Let’s consider a thin ribbon configuration with 4-unit cells along the edge and $4 \times M$ unit cells towards the other direction. By simulating the nonlinear evolution in this system, we see that prethermalization persists for longer propagation lengths as we increase M . This is due to the edge state localization effect which causes a decrease in the value of the overlap integrals between the bulk and edge states as the bulk becomes larger. To ensure a fair comparison between different M sizes, power was also appropriately scaled. As general rule of thumb, as a system doubles in size, it will generally need twice the power for similar thermal relaxation times. These results suggest that as M goes to infinity, ultimately, the edge states will have a vanishing contribution, as the reviewer correctly indicated. To clarify this issue, we have added a new figure in the supplementary (Fig. S3). We thank our reviewer for giving us the opportunity to elucidate these aspects.

Reviewer's comments: The authors need to explain qualitatively why their model predicts a non-zero bulk current. All bulk modes are symmetric with respect to positive and negative current directions, and therefore for any temperature the average bulk current can be expected to be zero. Probably this is some not very obvious feature of the system or of the model that I do not understand. In the electronic systems, the current in the bulk Landau levels is zero. Why is it different here? Why there is some negative contribution into the current from the bulk modes along the edges in Fig. 2?

Response: The reviewer is right, indeed, the net current from bulk contributions over the entire lattice structure will be net 0. Yet, what we study here is the thermalized edge current flowing at specific channels of the topological lattice. That is why this quantity is finite while when we consider the sum of these edge currents is of course zero. Nevertheless, there is a subtle difference when dealing with a ribbon geometry or a finite triangular topological lattice. This difference emerges from the number of channels involved in edge current transport.

In the ribbon case the currents associated with the fundamental bulk mode (highest-order mode) circulate in small paths, resulting into a 0 net average current (Fig. S2c). At the edges (same figure), this circulation leads to a small unidirectional current which flows in opposite directions between the bottom and top edge. The incoherent summation of all currents from all bulk modes is shown in Fig. S2d. In this case, we see that there are surprisingly zero bulk currents while at the edge channels there is a finite flow with a definite sign and direction. Nonetheless, the net current remains 0 as the top edge channel is balanced by the bottom flow. In Fig. S2e we show that, as expected, the total average current associated with the top edge states flows in the left direction. Similarly, in Fig. S2f the bottom edge states result in currents flowing in the right direction. The summation of the currents from Fig. S2d, Fig. S2e and Fig. S2f gives a net 0 current, both locally and globally. This is why at infinite temperatures where a statistical equipartition of bulk and edge modes occurs, the total edge currents will be zero ($J_B + J_E = 0$). For different temperatures, an imbalance between bulk, top and bottom edge state contributions results in finite net currents.

In the triangular configuration we follow a similar approach in analyzing these currents. In this case there is no bottom and top edge, all edge states are associated with a clockwise circulating net flow. On the other hand, all bulk states are associated with a counterclockwise net flow at the edges (similarly to the comparison between Figs. S2d, e and f). The average net contribution of either J_B or J_E will be always net 0 because the structure is now finite (the current cannot escape). Instead, we analyze the net current circulating in the triangular structure by calculating the projection of the currents along the edges of the lattice (in this case in the clockwise direction). These results are shown in Fig.2b (of the main manuscript) for different temperatures. To clarify these points, we added a new section at the Supplementary Material (please see Section IV).

Fig. S2(a) The Haldane lattice with 160 sites (b) The band structure with parameters ($\Delta = 0, t_1 = 1, t_2 = 0.2, \varphi = \frac{\pi}{2}$) in a ribbon configuration with periodic boundary conditions at the armchair edges, (c) local currents for the fundamental (highest-order) mode. (d-f) Total current contributed by (d) the bulk modes (e) the top edge states (f) the bottom edge states at modal equipartition (g) The local currents at the edge site marked in the red (a) and (h) the local currents at the bulk site marked in green in (a).

Reviewer's comments: In the supplementary, the authors state that “during prethermalization, the two bands exchange optical energy much faster than power”. Are there any explanations or arguments in favor of this statement?

Response: We have consistently observed this behavior in all of our simulations. Generally, the exchange of energy and power is dictated by the way the states or modes are nonlinearly coupled to each other. For a relatively wide bandgap, the phase matching between modes in the same band group allows for stronger intraband coupling. On the other hand, for interband exchanges we know that the four-wave mixing process are primarily responsible for exchanging the optical power while the cross-phase modulation governs the exchange of energy between the two bands. In our case, the former are weaker than the latter because the cross-phase modulation process is always self-phase matched while the one

associated with the 4-wave mixing is not. More specifically, 4-wave mixing is more prone not only to phase mismatching conditions but also to intensity fluctuations resulting from self-phase and cross-phase modulation effects. This explains why in our system energy exchange happens faster than power transfer between the bands of this topological system.

Reviewer's comments: Some small typos should be corrected, including the word « eignevalue » in Fig 1.

This typo has now been corrected. We thank the reviewer for her/his observations.

Reviewer 2

We would like to thank our reviewer for finding our results interesting and for all the constructive comments and suggestions. We hope that the reviewer will now find our revised manuscript acceptable for publication in Nature Communication.

Reviewer's comments: Very recently, the formation of Floquet solitons and quantised nonlinear transport has been experimentally observed in topological lattices, see Science 368, 856 (2020) and Nature 596, 63 (2021) [The authors may cite these papers].

Response: We would like to thank the reviewer for bringing these works to our attention. We now cite these papers in the revised manuscript.

Reviewer's comments: I believe that this work deserves publication in a decent journal. However, I think a general reader would like to know about the challenges in experimentally probing such effects. Apparently, probing optical states for an extremely long evolution time (or propagation distance) is difficult using photonic topological lattices. Additionally, both periodic and open boundary conditions are used for analysing the thermalisation process -- it would be nice to highlight how the periodic boundary condition can be realised in experiments.

Response: We would like to thank our reviewer for bringing this aspect up. Quite recently, optical thermalization effects have been observed for the first time in optical multimode fibers (Nat. Phys. (2022) doi:10.1038/s41567-022-01579-y and Opt. Express 30, 10850 (2022)) and time-synthetic lattices (A. L. M. Muniz et al., "Optical Thermodynamics in Nonlinear Photonic Lattices," 2020 Conference on Lasers and Electro-Optics (CLEO), 2020, pp. 1-2.). We now cite these papers given that at the time of the submission of our paper, these articles were not yet published. In principle photonic topological lattices can be inscribed in glass (Ref. 9,53,54) or can be implemented in time-synthetic dimensions. Given the long evolution times required to observe thermalization effects, a possible platform to observe such thermalization phenomena in 2D nonlinear topological lattices could be time-synthetic arrangements along the lines discussed in Refs 55,56. These 2D systems are typically implemented involving 4 fiber loops and allow more than 600 exchange times - that is more than enough to observe effects along the lines discussed in our paper. In general, ribbon like geometries with periodic boundary conditions can also be implemented in 2D time-synthetic lattices.

Reviewer's comments: Given the high impact of the journal, it would be nice to elaborate further on the most significant aspect of their results and the perspectives for new research that will be enabled by it.

Response: We agree with our reviewer that an elaboration on the most significant aspects of our work and on future research directions will be beneficial to the readership. We have now included these aspects in the Discussion section of our revised manuscript. Please see pages 7-8. We hope that our reviewer will now find these amendments productive.

Reviewer 3

We would like to thank our reviewer for finding our work interesting and relevant to the community working in the field of topological physics. Moreover, we would also like to thank our reviewer for his/her useful comments.

Reviewer's comments: 1. In the text, on page 6, the authors state that at $T \rightarrow \infty$ the currents J_E and J_B "negate each other resulting in a net zero average current". However, I do not see this in Fig. 2a where J_T is not approaching zero for large t . Moreover, J_E (red) + J_B (blue) is not equal to J_T (green) in Fig. 2a.

Response: The infinite temperature case is shown in the top-right panel of Fig.2b where indeed the net average current is zero. In Fig.2a we instead show results corresponding to the thermal evolution at a finite temperature, $T=0.19$. We added a label to indicate this in order to avoid any confusion. The reviewer is right that there was a problem in our previous figure in Fig2. a. We apologize for this mistake resulting from an incorrect scaling used. We have rectified this issue by using the correct scaling in which case J_E (red) + J_B (blue) is equal to J_T (green). We thank our reviewer for bringing this issue up.

Reviewer's comments: 2. How did the authors separate edge modes from bulk modes in the first place? Figures 1c and 1d suggest that they consider edge modes as the modes for energies where - without edges - there would be the band gap. On the other hand, one should look at the localization of the modes close to the edges, and then, by definition, there can be no contribution from bulk modes to the edge currents. The authors should clarify their choice.

Response: Indeed, the approach we took to identify the edge states was through the observation of edge localization. However, in a honeycomb lattice with no sublattice detuning ($\Delta = 0$), as in the cases in Fig 1c and 1d, all edge states reside within the band gap so the two definitions are equivalent. In the case where $\Delta \neq 0$, one has to identify the edge states of the lattice after computation by observing their corresponding edge localization.

As to bulk modes, one can indeed expect the net current to be zero. However, for the periodic ribbon configuration, one can find a non-zero local contribution to the currents at each separate edge channel, flowing in opposite directions. To visualize this, we added a new figure and a new section in the Supplementary Material (please see Section IV). There, we analyze the origin of the bulk and edge state contributions for a ribbon configuration. For example, we see that in the case of an equipartition of bulk and edge modes (at an infinite temperature state) the incoherent summation of bulk-originated currents (J_B) reveals unidirectional current channels at the boundaries that counteract the edge state contributions. This leads to a net 0 flow (in both the triangular and ribbon configuration).

In the triangular configuration there is only a single channel and in our case all edge states are associated with a clockwise circulating net flow. The average net contribution of either J_B or J_E will be indeed net 0 because the structure is finite (the current cannot escape). Instead, we analyze the net current circulating in the structure by calculating the projection of the currents along the edges of the lattice (in this case in

the clockwise direction). These results are shown in Fig.2b (of the main manuscript) for different temperatures.

Reviewer's comments: 3. *I had a hard time to go through (and understand) all panels in Fig. 3. First of all: Is the time axis in 3b, lower right plot, in the right direction? $T=\infty$ is at $t=0$ there. Further, in the caption, the authors write for the upper plot in 3a about " $\gamma=0$ and $\gamma=2$, respectively". Hence, I first was looking for two curves until I understood that one moves along the theoretical, red line for nonvanishing γ and does not for $\gamma=0$. Further, the authors write that "the initial and equilibrium modal occupancies" are depicted in the middle panel. It is clear that the lines are the equilibrium distributions. The fact that the shaded areas indicate the initial occupancies follows from what is written in the main text but not from the caption. The authors should improve on the caption and/or the figure itself.*

Response: We would like to thank our reviewer for suggesting several potential improvements for Fig. 3. Indeed, the time axis in 3b was in the wrong direction. This is now corrected in the revised document. Considering the difficulty our reviewer expressed in understanding Fig.3a we made an effort to clarify some aspects. The figure and caption are now improved by taking into account all points raised by our reviewer.

Reviewer's comments: 4. *There is a small phi in Fig. 2b at the axes but a capital phi in the text.*

Response: We thank our reviewer for pointing this out. We have rectified his issue in the revised manuscript.

Reviewer's comments: 5. *It should read "loss of the internal energy" in the figure caption of Fig. 3, not "lose of the internal energy".*

Response: This typo has now been corrected. We thank our reviewer for bringing this to our attention.

REVIEWERS' COMMENTS

Reviewer #1 (Remarks to the Author):

The manuscript has become much clearer. The central results of the study are easier to get, and the conditions at which they are obtained are also given more explicitly. I thank the authors for their efforts.

I think that this work really merits to be published in Nature Communications. The existence of a chiral edge current in topological systems even in presence of interactions is an important result, and the thermodynamical description of such systems is an important step forward, making another bridge with fermionic topological systems. This is an important interdisciplinary aspect of this work.

One more typo to be corrected: line 224, "reviles" should be replaced by "reveals". Moreover, I think that the authors should double-check everything. They write that they have corrected the typo in Fig. 1, but unfortunately the word "eignevalue" is still there. Is it the wrong version of the Figure?

Reviewer #3 (Remarks to the Author):

I am fully satisfied with the authors' response to my critique and the revised manuscript. I recommend now publication in Nature Communications.